# Wise Information Technology of Med: Human Pose Recognition in Elderly Care

**DOI:** 10.3390/s21217130

**Published:** 2021-10-27

**Authors:** Difei Xu, Xuelei Qi, Chen Li, Ziheng Sheng, Hailong Huang

**Affiliations:** 1School of Electrical Engineering and Telecommunications, University of New South Wales, Sydney, NSW 2052, Australia; z5150990@zmail.unsw.edu.au (D.X.); chen.li1406@unsw.edu.au (C.L.); ziheng.sheng@unsw.edu.au (Z.S.); 2College of Information Science and Engineering, Northeastern University, Shenyang 110819, China; qixuelei@163.com; 3School of Civil and Environmental Engineering, University of New South Wales, Sydney, NSW 2052, Australia; 4Department of Aeronautical and Aviation Engineering, The Hong Kong Polytechnic University, Hong Kong

**Keywords:** elderly care, human pose recognition, feature extraction, PCA-LSTM recognition, gaussian kernel function classification

## Abstract

The growing problem of aging has led to a social concern on how to take care of the elderly living alone. Many traditional methods based on visual cameras have been used in elder monitoring. However, these methods are difficult to be applied in daily life, limited by high storage space with the camera, low-speed information processing, sensitivity to lighting, the blind area in vision, and the possibility of revealing privacy. Therefore, wise information technology of the Med System based on the micro-Doppler effect and Ultra Wide Band (UWB) radar for human pose recognition in the elderly living alone is proposed to effectively identify and classify the human poses in static and moving conditions. In recognition processing, an improved PCA-LSTM approach is proposed by combing with the Principal Component Analysis (PCA) and Long Short Term Memory (LSTM) to integrate the micro-Doppler features and time sequence of the human body to classify and recognize the human postures. Moreover, the classification accuracy with different kernel functions in the Support Vector Machine (SVM) is also studied. In the real experiment, there are two healthy men and one woman (22–26 years old) selected to imitate the movements of the elderly and slowly perform five postures (from sitting to standing, from standing to sitting, walking in place, falling and boxing). The experimental results show that the resolution of the entire system for the five actions reaches 99.1% in the case of using Gaussian kernel function, so the proposed method is effective and the Gaussian kernel function is suitable for human pose recognition.

## 1. Introduction

The economic consequences of an aging population published by the United Nations in 1956 established the population classification standard. The number of people aged 65 and above in a country or region is more than 7% of the total population [1]. Japan is a country with serious aging, and its population has been decreasing for nine consecutive years. According to the statistics in 2019, the population over 65 years old in Japan is 35.885 million (28.4%) [2]. According to the data released by China’s National Bureau of Statistics, by the end of 2019, the number of older adults aged 60 and above will reach 254 million (18.1%), and the number of older people aged 65 and above will reach 176 million (12.6%) [3].

Therefore, in response to the increasingly serious aging trend, people have paid more attention to monitoring older adults living alone. These elderly people usually take dangerous actions (such as a sudden fall) that damage their health. The existing methods can be divided into wearable and non-contact. The common types of wearable sensors are bracelet and ankle monitor [4], which needs to be worn close to the body, and there are some problems, such as low universality, poor portability, high damage rate, single function, and high false alarm rate. As a traditional non-contact sensor, the camera has high storage space requirements and information processing ability and is sensitive to light and other conditions. Both camera and radar observation methods are developed based on non-wearable devices. Compared with cameras, UWB radar is not restricted by blocking objects. It can still play a monitoring role outside the visible range of the camera. In addition, the radar is not affected by light conditions and can still work in dark environments. This feature can monitor the nocturnal activities of the elderly [5]. Therefore, we propose a non-contact monitoring device based on UWB radar, which can effectively solve dim light and blind area problems.

UWB radar uses the Doppler effect to recognize the human posture by the mutual movement between the human self and the radar. Ref. [6] used continuous wave radar to extract Doppler features and a trained SVM to recognize seven rhythmic human movements. The accuracy of classification results is around 92.8%. Ref. [7] applied the Doppler radar to evaluate the Doppler characteristics of human walking, which shows that this method effectively recognizes the moving human body’s falling posture. In Ref. [8], the Dynamic Range Doppler Trajectory (DRDT) of human motion was extracted using frequency modulated continuous wave radar. Simultaneously, the DRDT was combined with the machine learning classification method to separate continuous motion into a separate motion, and it shows that the average classification accuracy is above 91.9%. In Ref. [9], a depth convolution neural network (DCNN) was applied to human activity classification based on micro-Doppler features, and it was transformed into an image classification problem. Ref. [10] used Doppler radar combined with the DCNN method and transfer learning method to carry out five kinds of human posture classification and recognition. Ref. [11] combined the micro-Doppler features with the DCNN method to recognize human hand posture. Based on the above analysis, human posture recognition based on radar can realize the classification and recognition of highly similar actions in a simple framework with high feasibility. Because of its high resolution, intense penetration, and low power consumption, UWB radar can effectively overcome the lighting and privacy problems of the camera, ensure all-weather work, and effectively solve the problem that the optical system does not work well due to occlusion. Thus, it acts as an essential role in the monitoring of the elderly.

We propose the PCA-LSTM algorithm. Principal Component Analysis (PCA) is a data analysis method. This algorithm transforms the original data into a set of linearly independent representations of each dimension through linear transformation, which can be used to extract the main feature components of the data. The PCA algorithm can reduce the dimensionality of the original complex data. As a result, it can greatly reduce the complexity of signal processing. Compared with other algorithms, PCA has the advantages of simplicity and no parameter restrictions. It can be used to extract the characteristics of the main information components of the signal, which can be used to filter out the noise contained in the signal. At the same time, it can also be used to combine signal characteristics. In the classification and recognition of human poses, similar poses will have some feature vectors with greater correlation. In this case, the PCA algorithm can merge the highly correlated features into one feature. When the information represented by the two features is the same, the algorithm can be used to filter out one of the redundant features. In addition, when there are many signal features and overfitting occurs, the PCA algorithm can preserve the really useful part of the signal. In order to prevent the features of similar actions from being too similar to make the feature vectors all integrated, the PCA algorithm is combined with Long Short-Term Memory (LSTM). The purpose is to solve the problem of gradient disappearance during long sequence training. The main function of LSTM is to control the transmission state through the gating state. LSTM can remember the information that needs to be remembered for a long time and forget the unimportant information. Although the Recurrent Neural Network (RNN) algorithm can also handle sequence-changing data, it only has a single memory stacking method compared with the LSTM algorithm, and it is not suitable for long-sequence memory tasks. When monitoring the elderly indoors, the PCA-LSTM algorithm can memorize the slow movements of the monitored person for a long time and accurately classify them. In addition, unimportant normal actions are merged and forgotten. Compared with other algorithms, the PCA-LSTM algorithm has few data processing and improved accuracy.

The main contributes of this paper are twofold. Firstly, an improved algorithm based on PCA-LSTM is proposed to integrate micro-Doppler features and time-sequence to recognize human posture. Secondly, the accuracy of different kernel functions in human posture classification is studied.

The rest of the paper is organized as follows. Section 2 discusses some closely related work. Section 3 presents the system model and formally states the considered problem. Section 4 presents the proposed algorithm with theoretical analysis, and Section 5 demonstrates the performance of this algorithm via simulations. Finally, Section 6 briefly concludes the paper.

## 2. Related Work

The human posture recognition system based on a UWB radar should include four essential parts: human echo data acquisition, echo data processing, feature extraction, and recognition algorithm (see Figure 1). Among them, echo data processing and feature extraction methods can be divided into three categories. The first method is based on traditional statistics, including statistical characteristics such as mean and variance of radar echo envelope. The second one is based on time-frequency transform, including Fourier transform and the short-time Fourier Transform (STFT), pseudo-Wigner distribution (PWD), and wavelet transform. The third one is based on component analysis, and it mainly includes the Principal Component Analysis (PCA) and the Independent Component Analysis (ICA).

Combined with the UWB radar’s characteristics, according to the different feature extraction methods, the human posture recognition method is divided into traditional machine learning and deep learning methods. Firstly, the traditional machine learning methods are mainly used in human posture recognition, such as SVM, k-Nearest Neighbour (kNN), and Random Forest (RF). These classification and recognition methods are mainly based on the statistical theory and shallow features extracted from the original echo data, such as mean, variance, Euclidean distance, Fourier transform [12]. The feature extraction methods, such as PCA, Discrete Cosine Transform (DCT), and the prominent shallow features are selected for recognition. Secondly, the method based on deep learning can effectively avoid the tedious manual feature extraction and selection in the traditional machine learning methods for UWB radar human posture recognition and improve performance. Compared with the traditional machine learning method, it can automatically learn the effective feature extraction mechanism through the data without manually designing the feature extraction process. In addition, it can also identify complex human activities while reducing the processing process. Ref. [13] proposed a classification and recognition algorithm based on a convolution neural network (CNN). The field of UWB radar human body recognition has attracted many scholars’ attention, and many derivative network models have been produced. Ref. [11] used UWB radar to collect micro-Doppler features of human gesture and then combined with an improved DCNN to recognize ten kinds of gesture. In the training phase, the 5-fold cross-validation method is used to improve the model’s generalization ability, and the average recognition rate is more than 90%, but it is still sensitive to the change of distance and azimuth. In order to solve this problem, [13,14] extracted three robust features from the pre-processed gesture echo signal and defines a fitting data algorithm, which detects the periodicity of gesture motion to eliminate the unexpected motion of hand or body, and achieves 95% accuracy.

For radar signal processing, researchers have proposed using time-frequency analysis to obtain the information in the signal. Fourier transform and its improvement methods are the main methods of time-frequency analysis. Ref. [15] reported the efficiency of the Fourier–Bessel transform and time-frequency-based method in conjunction with the fractional Fourier transform. The short-time Fourier transform processes the signal by selecting different window functions. This method can determine the frequency and phase of the sine wave in the time-varying signal’s local area [16,17,18]. Wavelet transform (WT) is also used in radar signal processing. Compared with the Fourier transform, this method’s transformed base is a wavelet base with limited length and attenuation [19,20,21]. Ref. [22] used WT to detect a person’s fall through a radar installed on the ceiling. The WT method uses wavelet decomposition coefficients of a given scale to determine the time location where falls may occur. The short-time Fourier transform (STFT) decomposes the entire time domain process into countless small processes of equal length, and each small process is approximately stable. After that, Fourier transform is performed on each small process. Compared with STFT, the wavelet transform replaces the infinitely long trigonometric function base with a finitely long attenuating wavelet base. This wavelet base can be translated and stretched so that the signal can be analyzed at different times and different frequency ranges. However, wavelet transform is not applicable in this situation. If wavelet transform is used to process the collected data, the choice of wavelet base and the choice of scale function are different in each action. In addition, after processing the echo data by using different wavelet bases and scale functions, the comparability of the obtained attitude features will be reduced. In contrast, since the posture of the elderly changes slowly, the signal in each window can be regarded as a relatively stable signal when using STFT. Furthermore, the feature vectors obtained by STFT are more comparable.

The classification method is used to classify different poses after obtaining the time-frequency information and micro-Doppler features of different human body movements. Researchers have developed a variety of classification methods for high-dimensional data. The KNN method is one of them. Ref. [23] used the KNN algorithm to classify poses based on the ratio and difference of the human contour bounding box. Hämäläinen M. et al. combined the KNN algorithm with UWB radar to detect the posture of the human body without camera surveillance. Furthermore, they also analyzed the reliability and fault tolerance of the UWB radar network framework. The experimental results show that the accuracy of the system can reach 99% for the static posture of the human body [24]. Besides, the SVM is also widely used in classification scenarios. Ref. [25] used SVM to classify heel acceleration and plantar pressure data to determine whether the human body is in a sedentary posture. Mizumoto T. et al. [26] combined the RF algorithm and Kinect sensor with a microbehavior sensing system. This system can identify microbehaviors by extracting features from data. The accuracy rate can reach 78%.

Currently, the comprehensive systems that integrates data collection, hardware improvements, and classification methods to classify human posture has attracted the attention of many researchers. Sizhe An et al. [27] developed an intelligent medical assistance system based on millimeter waves. The main purpose of the system is to allow those with movement disorders to recover from sports. In rehabilitation treatment, patients are required to perform standard actions to achieve the purpose of rehabilitation. The judgment of whether the movement is standard or not requires very precise joint positioning to be realized. However, the high-precision action judgment increases the complexity and manufacturing cost of this system. Furthermore, the system used in this article needs to be additionally equipped with Nvidia Jetson Xavier-NX to work. This is inconsistent with the purpose of monitoring the elderly. Sengupta A et al. [28] tracked human bones. The authors used the reflected signals of millimeter wave radar to detect 15 different bone joints. The main actions proposed are focused on the changes in the bones of the upper limbs of the human body, such as the swinging of the arms, and the CNN algorithm is used to structure the image according to the information of the radar echo. However, this article focuses on the change of the overall posture of the human body. When monitoring the elderly, the overall posture of the human body is not judged by the changes of several joints. Xue H et al. [29] used millimeter-wave radar and motion capture technology to perform 3D real-time modeling of the human body’s active posture. The posture of the human body obtained in this article is very delicate. In addition, it can be presented on the computer in real time through a motion capture system. However, the VICON motion capture system used in the article requires huge space and requires the assistance of a camera, so it is not suitable for home use.

## 3. System Model and Problem Statement

Most human pose recognition and classification methods use cameras and optical sensors to collect data to determine the real-time human poses. However, dark light and long acquisition distance lead to inaccurate fusion data. Radar has the advantages of fewer restrictions and high accuracy in acquiring human posture data. As a result, there is no need to attach other equipment to the human body to collect real-time posture data. In addition, cameras collect data in the form of images or videos. Compared with radar, cameras are more likely to leak the privacy of the monitored person. Radar collects echo data directly, so privacy and confidentiality are better. We propose a human posture recognition method based on UWB radar, and the structure of the proposed system is shown in Figure 2:

1. UWB radar is responsible for collecting human posture data, including: from standing to sitting, from sitting to standing, walking in place, falling, periodic boxing. The dynamic motion of the human body relative to radar produces the Doppler effect. The echo data collected by radar is converted into time-frequency data by STFT, which reflects the movement characteristics of the human body.

2. Six motion features of the human body are extracted according to the Doppler shift and micro-Doppler features of the human body, and these six features are saved as vector form. The system classifies the human pose and studies the classification and selection of different kernel functions in SVM. As a result, this system can obtain the appropriate kernel function for indoor human posture recognition.

3. The collected data and data sets are trained to evaluate the accuracy of recognition and classification.

When the signal is transmitted to the target, a part of the energy of the signal will be reflected by the scatterer constituting the target and be observed by radar. Joseph Keller [30] believes that when the operating frequency of the radar is high enough, the scattering behavior of complex targets can be succinctly modeled as the sum of the scattering responses based on a simple scattering mechanism. Therefore, the physical correlation model of the measured radar echo $S\left(\omega ,k\right)$ can be constructed [31].
(1)$$S\left(\omega ,k\right)=\sum_{m=1}^{N_{S}}S_{\Gamma_{m}}\left(\omega ,k;\theta_{m}\left(k\right)\right)e^{-j\frac{4\pi \omega }{c}r_{m}\left(k\right)}+v\left(\omega ,k\right)$$
where $N_{S}$ is the order of the model; $S_{\Gamma_{m}}\left(.\right)$ is the scattering behavior model of type $\Gamma_{m}$, which is a function of the sampling frequency ω and the number of pulses k and has a parameter $\theta_{m}\left(k\right)$. During the pulse k, the distance of the m-th scatterer is $r_{m}\left(k\right)$, c is the speed of light, and $v\left(\omega ,k\right)$ is the additional white Gaussian noise. The observation result is the accumulation of $k=1,2,3,\cdots ,N_{k}$ pulses and $n=1,2,3,\cdots ,N_{F}$ radar echo signals with frequencies $\omega_{n}$. When the object characteristics of the target are much smaller than the resolution of the radar, it is sufficient to limit the above equation to the sum of point type scatterers. Without considering the inaccuracy and unambiguously estimated free variables, the scattering behavior model can be simplified as $S_{\Gamma_{m}}\left(\omega ,k;\theta_{m}\left(k\right)\right)=A_{m}\left(k\right)$. Thus,
(2)$$S\left(\omega ,k\right)=A_{m}\left(k\right)e^{-j\frac{4\pi \omega }{c}r_{m}\left(k\right)}+v\left(\omega ,k\right)$$
where $A_{m}\left(k\right)$ is the scattering coefficient of the m-th feature on pulse k. Therefore, the human body can be regarded as many point scatterers propagating in space. Through the STFT of the observed radar echo signal, the distance can be directly connected with the Doppler effect.

When the human body moves, the UWB radar receives the echo signal, which contains the motion information of the human body, in which the Doppler shift reflects the speed of human movement. Based on the time-frequency analysis of the echo signal of human motion, the micro-Doppler characteristic data are obtained. The normalized energy, variance, skewness, and kurtosis of the radar echo of human motion are used to distinguish human motion behaviors: from sitting to standing and from standing to sitting. We mainly use the PCA method to reduce dimensions according to the mean value, variance, and speed of PCA coefficients and recognize human actions: walking, falling, sitting, and boxing. The following six micro-Doppler features is used to express human motion posture, including: Torso Doppler Frequency, Total Bandwidth (BW) of the Doppler Signal, Total Doppler Signal Offset, Torso Doppler Frequency Offset, BW of the Doppler Signal Without Offset and Standard Deviation (STD) of Normalized Doppler Signal Strength, in which Torso Doppler Frequency is the average frequency corresponding to the maximum energy of each frame in the radar echo signal and Total BW of Doppler Signal represents the movement of the human body. The highest and lowest frequencies of each time window constitute the high-frequency and low-frequency envelopes. It is the average difference between the low-frequency envelope’s minimum frequency and the high-frequency envelope’s maximum frequency. Total Doppler Signal Offset is the mean of the minimum frequency of the low-frequency envelope and the maximum frequency of the high-frequency envelope. Torso Doppler Frequency Offset is the average of the maximum and minimum trunk frequency. Bandwidth of Doppler Signal Without Offset is the average difference between the maximum frequency of the low-frequency envelope and the minimum frequency of the high-frequency envelope, which can describe the up and downswing of the human torso. Standard Deviation (STD) normalized Doppler Signal Strength represents the interaction between human motion and micro-Doppler radar.

Because of human motion signals’ non-stationary, non-linear and instantaneous characteristics, it is not easy to obtain the parameters that can effectively describe and distinguish the complex and changeable motion signals by simple time-frequency transformation. Therefore, the second feature is mainly extracted from the time spectrum through the time-frequency analysis of the radar echo signal of human motion. For example, the spectral features of STFT are used for motion state recognition.

In general, there are many kinds of motion signal eigenvalues. Different extraction methods and complexity will lead to different operation times. In addition, the classification accuracy changes with the feature selection. Therefore, in classifying human motion state recognition, it is of great significance to select the appropriate feature value according to the motion signal’s characteristics to improve the accuracy of recognition and classification.

Problem Statement: Aiming at the problems of low recognition rate when using time-frequency analysis in human pose recognition, the problems considered are as follows: (1) for the PCA method, the feature extraction needs to be improved; (2) for the SVM classification model, the kernel function needs to be selected suitably for human pose recognition.

## 4. Proposed Recognition and Classification Method

### 4.1. Recognition Algorithm (PCA-LSTM)

In traditional recognition methods, the PCA method plays an important role in shallow feature recognition, but the success rate of recognition is low. In Ref. [32], the energy distribution characteristics of UWB radar signals of eight human postures are extracted by wavelet packet decomposition method, and the parameters *C* and σ of SVM are optimized by using improved chaos adaptive genetic algorithm (ICAGA). The recognition rate can reach 97.6%. In Ref. [33], the concept of feature energy based on PCA and DCT is proposed. In the model training stage, the parameters *C* and σ are optimized by using the grid search algorithm (GS). It can be shown in final verification that when $\left(C,\sigma \right)\in \left(2^{-5},25\right)$, the recognition rate reaches around 96.09%; when $\left(C,\sigma \right)\in \left(2^{-8},28\right)$, the rate is about 98.04%, and the average rate is above 96%. Human pose recognition based on the traditional machine learning method extracts many shallow features, but it is still a problem whether the extracted features can ultimately benefit recognition. The diversity of shallow features will potentially increase the redundancy between different features and reduce the recognition accuracy. Therefore, the key points are the analysis of echo signals, feature extraction, and feature selection.

In order to combine the advantages of LSTM and PCA methods, we propose a dual-channel feature extraction model of LSTM-PCA, a human posture recognition method based on a dual-channel feature extraction model of the LSTM model. The PCA features combined with time cycle characteristics cast an essential role in improving human posture recognition accuracy. The period parameters are added to the original six features, which can extract the spatiotemporal features of human posture and complete the recognition of human posture. As shown in Figure 3, the PCA-LSTM dual-channel human posture recognition fusion model mainly includes three parts: the first part is composed of PCA, mainly used for extracting six features of human posture; the second part is composed of LSTM, which is mainly used to extract the radar data sequence features; the third part is feature fusion. SVM, PCA, and LSTM theory are combined to extract features and classify human posture. Therefore, a new micro-Doppler feature Time Sequence is added into the system.

### 4.2. Classification Algorithm

Although Doppler features can identify human motion features with apparent differences, approximate differences cannot be distinguished. This paper proposes a method to classify six human activities based on SVM to calculate Doppler features. Support vector machine uses the kernel function to transform sampling data into high-dimensional space. It optimizes the training data according to the support features. Next, SVM generates the corresponding model according to the known class name. The structure of SVM is shown in Figure 4.

Assume that there is a data set $\left[\left(x_{i},y_{i}\right)\right],x_{i}\in R,i\in N,y_{i}\in \left(-1,1\right)$, where $y_{i}$ is the output of the system, and $x_{i}$ can be linearized in $\omega^{T}x_{i}+b=0$, ω is the weight vector, *b* is the optimal classification surface offset. Therefore, by constructing the optimal classification surface of the original sample space, it can be expressed as a kind of constrains with $\min_{\omega ,b}\frac{1}{2}\omega^{T}\omega$ in (3).
(3)$$\begin{array}{c} \min_{\omega ,b}\left(\theta \left(\omega \right)\right)=\min_{\omega ,b}\left(\frac{1}{2}\omega^{T}\omega \right)\\ s.t.y_{i}\left(\omega^{T}x_{i}+b=0\right)-1\geq 0 \end{array}$$

If the original input space is transformed into a new high-dimensional feature space by introducing a kernel function $k\left(x,x_{i}\right)$ in linearly nonseparable, a linear classification hyperplane is constructed in the new feature space. This characteristic space is determined by $\phi \left(x\right)$, and its inner product operation is replaced by kernel function, that is, $k\left(x,x_{i}\right)=\langle \phi \left(x\right),\phi \left(x_{i}\right)\rangle$.

Some sample data points will deviate from the normal position in the new feature space and can not meet the constraint conditions. At this time, we can introduce a penalty factor *C* and add a relaxation term $\xi_{i}$ to relax the constraint conditions of linear separability so that the linear non-separability becomes linear separability. Therefore, (3) can be rewritten as (4). Moreover, (4) is a convex quadratic optimization problem. In order to solve this problem, the Lagrange function of this optimization problem is defined in (5).
(4)$$\begin{array}{c} \min_{\omega ,b}\left(\theta \left(\omega \right)\right)=\min_{\omega ,b}\left(\frac{1}{2}\omega^{T}\omega +C\sum_{i=1}^{N}\xi_{i}\right)\\ s.t.y_{i}\left(\omega^{T}x_{i}+b=0\right)+\xi_{i}\geq 1 \end{array}$$
(5)$$\begin{array}{c} L\left(\omega ,b,\xi ,\alpha ,u\right)=\frac{1}{2}\omega^{T}\omega +C\sum_{i=1}^{N}\xi_{i}\\ -\sum_{i=1}^{N}\alpha_{i}\left[y_{i}\left(\omega^{T}x_{i}+b\right)-1+\xi_{i}\right]\\ s.t.\alpha_{i},u_{i}\geq 0 \end{array}$$
where $a_{i}$ denotes the Lagrange multipliers and it is positive definite. By solving the dual problem equivalent to the original problem, the optimal solution of the original problem is obtained in (6).
(6)$$\begin{array}{c} E\left(\alpha \right)=\sum_{i=1}^{N}\alpha_{i}-\frac{1}{2}\sum_{i=1,j=1}^{N}\alpha_{i}\alpha_{j}y_{i}y_{j}\left(x_{i}x_{j}\right)\\ s.t.\sum_{i=1}^{N}\alpha_{i}y_{i}=0\\ s.t.0\leq \alpha_{i}\leq C \end{array}$$
where (6) can transform the linear non-separable problem into a linear separable problem, which only needs the maximum value of solution $E\left(\alpha \right)$. On the premise of choosing the appropriate value of the kernel parameter σ, increasing the penalty factor *C* can make the separability of the data tend to be stable, otherwise the generalization ability will be reduced. Therefore, the selection of kernel parameter σ and penalty factor *C* is significant.

### 4.3. Kernel Function in SVM

The human pose recognition method focuses on the improvement of feature extraction and optimization of the SVM model. The kernel function plays a significant role in SVM classification, so it is necessary to study different kernel functions. The kernel function is used to measure the similarity between samples. The expressions of several standard kernel functions are given in Table 1.

## 5. Experiment Results and Analysis

### 5.1. Radar and Scene Settings

The XeThru-X4 radar is used as a compact on-chip pulse radio UWB radar. The radar sends out electromagnetic pulses through the Tx antenna, and the electromagnetic pulses will be reflected from any object in front. As a result, reflective objects will propagate backward. The ultra-wideband radar model used in this experiment is XeThru X4M03. It is a baseband narrow pulse radar (IR-UWB). This type of ultra-wideband radar uses the module communication protocol wrapper protocol. MCP Wrapper goes one step further for embedded host implementation adding a wrapper with convenience methods around the MCP. The pulse is set to 15.1875 MHz, and the radar elevation is 2 m. The range of digital to analog converter (DAC) is set to 949–1100. Simultaneously, the number of iterations of the radar repeat DAC scan is set to 64.

It can be seen from Figure 5 that the test scene is built and attached to the absorbing materials to isolate the environmental noise. The training data set is a total of 225 sets of data. There are three testers, each of whom did 15 sets of actions. Informed consent was obtained from all subjects involved in the study. The test data set consists of three people doing five sets of each action, a total of 75 sets of testing data. Each tester stood 1.5 m directly in front of the radar. The specific situation is analyzed and compared at the end of the experiment. The five movements selected in this article are from sitting to standing, from standing to sitting, walking in place, falling, and boxing. In the first classification, they are divided into three groups according to the complexity of the action. The three sets of human postures are compared separately according to radar echo signals in the following experiment. Standing and sitting are the most basic postures of the human body. The elderly are more likely to fall when performing this set of postures. Boxing and walking in place are manifestations of the elderly who are sick or in need of help. This set of postures can help monitors to distinguish whether the elderly are in a dangerous. The schematic diagram of data collection is shown in the figure below.

### 5.2. Human Pose Recognition Analysis

This subsection mainly compares and analyzes the recognition of human posture. The comparative analysis between two types of data sets, including public data set based on vision and test data set based on UWB radar. The data set based on vision is rich, including simple actions to complex actions, and the sample size is large enough. Compared with the visual data set, the open data set based on UWB radar is less. Therefore, in the actual experiment, the UWB radar collected 225 groups of human movements. Figure 6 compares the human posture recognition rate performance in different data sets and compares the sample size in different data sets. It can be intuitively concluded that the current research on human posture recognition using UWB radar is based on self-test data sets, and the recognition rate based on time-frequency analysis is relatively high, with an average of more than 85%. In addition, the sample size based on vision is relatively large, but the sample size of human pose recognition based on UWB radar is generally small.

### 5.3. Human Pose Classification Analysis

After setting up the radar and experimental environment, it can collect pose data from the radar. There are three groups to analyze and make the comparison: the first group is the action from sitting to standing and the action from standing to sitting; the second group is walking in place and falling actions; finally, the third group is the boxing action. Each action can be divided into six processing, including: (a) the original data, (b) frequency spectrum diagram before noise reduction, (c) frequency spectrum diagram after noise reduction, (d) action interval, (e) body frequency, (f) compensation.

Now, we analyze the first scenario: sitting and standing in Figure 7 and Figure 8. It mainly lies in the time interval of action and the direction of trunk movement. Observe whether it is close to radar or far away from the radar. Furthermore, the movement of the tester in the test should be adequately described.

When classifying the two actions from sitting to standing and from standing to sitting, some differences can be seen from the time-frequency analysis graph. Firstly, the time difference between the rise and fall of the waveform is shown from the original radar echo data (Figure 7a and Figure 8a). Figure 7a shows that the waveform’s peak-to-peak value starts to increase after the start of detection, then begins to decrease after reaching the maximum value, and finally stabilizes and fluctuates within a specific range. Compared with standing, in Figure 8a, the waveform remains stable initially, and then the peak-to-peak value of the waveform gradually increases until it reaches the maximum and stays near the maximum. Secondly, in the spectrogram (Figure 7c and Figure 8c) after noise reduction, the difference between the two actions is also apparent. In Figure 7c, the frequency is mainly concentrated in the time window 550 ms position, while in Figure 8c, the frequency is mainly concentrated in the 100 ms position. Besides, in the torso frequency diagrams of the two actions (Figure 7e and Figure 8e), the rate of change of the torso from bending to the upright in the second half of the action is the largest when the sitting action occurs. The torso frequency peak occurs in the second half of the time window. In contrast, when the standing action occurs, the torso frequency peak is in the first half of the waveform.

Then, walking in place and falling become the second scenarios in Figure 9 and Figure 10.

The original radar echo data (Figure 9a) shows that walking in place is a periodic action, and the waveform of the radar echo shows prominent periodic characteristics. However, the instantaneous falling action makes a significant fluctuation in the radar echo’s original waveform (Figure 10a). Signal rapidly decreases to reach a steady state after a peak quickly. Besides, the movement occurrence interval (Figure 9d) also shows the periodic characteristic of walking in place. In contrast, in the falling motion occurrence interval (Figure 10d), the motion only occurs in 140 ms to 240 ms. Furthermore, the spectrogram of walking in place (Figure 9c) shows that the action period is about 150 ms, and the frequency distribution is uniform. However, the frequency is concentrated between 140 ms and 240 ms, and the action is instantaneous in the spectrogram of falling, shown in Figure 10c. In addition, due to the periodic movement of the torso that is walking in place, the torso’s frequency also changes periodically with the body swinging back and forth (Figure 9e). The fall action’s torso frequency shows a drastic change in the process from leaning the body to lie on the ground and then gradually returns to a stable state (Figure 10e).

Now, the third scenario is the boxing action in Figure 11. The boxing action is periodic. The action occurrence interval is shown in Figure 11d, and the raw radar echo data can be shown in Figure 11a. However, the spectrogram after noise reduction reflects in Figure 11c, in which each action cycle shows the characteristics of a large frequency concentration interval. This is because the arm will rotate by itself during the having action, and multiple frequencies will be superimposed. The movement situation is simple. This makes the frequency concentration interval in the spectrogram of the boxing action less. Furthermore, in Figure 11e, the torso frequency of boxing changes faster. This is because boxing is complicated and involves many movements of the torso.

### 5.4. Kernel Function Comparison Analysis

There are 225 sets of test data substituted into the model for classification. Let define 0 to 4 be standing, sitting, walking in place, falling, and boxing, respectively.

In the results of different classification methods, the substituted data are the feature vector of each group of actions. The abscissas and ordinates have no specific physical meaning, and the numbers only represent the scale of the feature vector in the classification process. Four different classification methods are used for data training and classification after obtaining five human body posture feature data. Figure 12a–d are the results of various classification methods. The shape of the points in each figure represents whether the classification is accurate. The dot indicates that the data classification result is consistent with the original data category it represents, and the fork point indicates that it does not match. Figure 12a–d show that both the decision tree and KNN classification methods have more fork points. The accuracy of data classification is not high. In contrast, Gaussian SVM and Gaussian Naive Bayes’ classification methods only have two sets of data errors. However, in Figure 12a, there are two fork points of yellow and blue. It shows that when the train data uses the Gaussian Naive Bayes classification approach, the same real class data are mistakenly divided into two prediction classes. Compared with Figure 12b, there are two yellow fork points in Figure 12c. It shows that fundamental data are erroneously divided into prediction data when using the Gaussian SVM method. Moreover, Table 2 shows the different classification accuracy in Decision Tree, Gaussian Bayes, Gaussian SVM and KNN. Therefore, the Gaussian SVM is stable, and it does not classify the same human posture into two different human postures.

When studying Gaussian SVM’s use to classify human pose information, we introduced linear SVM classification methods for comparison. It can be seen from Figure 13a that the linear SVM classifies the same kind of real data into two different prediction data, and the linear SVM is not as good as Gaussian SVM for the classification of human body pose data. However, the Gaussian SVM has different classification results due to different kernel scale choices in the kernel function. In Figure 13b, when the kernel scale is selected to be small, over-fitting is prone to occur. The support vector sample’s effect increases, but the unknown sample classification effect is inferior because there are multiple fork points in Figure 13c. However, when the kernel scale is large, under-fitting is prone to occur. SVM can classify sample data into the same type of data, which results in a set of data in Figure 11 that was initially purple (walking in place) was incorrectly classified as yellow (boxing). Compared with the above two nuclear scales’ values, an appropriate kernel scale selected in Figure 13d can more accurately classify all data using Gaussian SVM. Moreover, Table 3 shows the Classification Comparison in SVM with Different Kernel Scale. The wrong data are classified incorrectly due to the tester’s action error, which is acceptable. The accuracy rate reached 99.1%.

## 6. Conclusions

In this experiment, the PCA-LSTM algorithm combined with the SVM classification method is used to classify and recognize the human posture. In order to prevent the elderly from physical damage during data collection, two healthy males and one female (22–26 years old) were selected in this experiment to perform five postures while imitating the slowness of the elderly. In the experiment, it was found that the movement speed of the measured object and the degree of similarity of the movement are the limitations of this experiment. In the data collection for similar postures from standing to sitting and from sitting to standing, the person under test will slow down the movement and increase the movement range of the body. The main bone joints on the leg are all driven and move at a very slow speed. However, this group of similar postures can still be distinguished according to the direction of movement of the torso and this result also proves that this system still has a certain degree of robustness under the disturbance of similar postures.

The most influential factor in this experiment is except for the action of the tested object. The position of the radar also has a greater impact on the data of this experiment. The radar is placed directly in front of the measured object and the height is at the position of the chest and abdomen of the human body. This height allows the UWB radar used this time to collect the information of the measured object’s movement to the greatest extent within its radiation angle range. Although the data collected when the angle of view deviates from the front of the radar are very similar to the results obtained from the data directly in front, the change of the angle causes the feature vector to change and needs to be retrained.

In future research, we will focus our research on the detection and classification of posture when the action changes slowly. As the elderly move slowly indoors, they will slowly fall to the ground and sit down when they have symptoms. At present, the detection of this situation can only be observed by the camera and cannot be determined by the radar. Research in this area will be the focus of our future research.

## Figures and Tables

**Figure 1 sensors-21-07130-f001:**
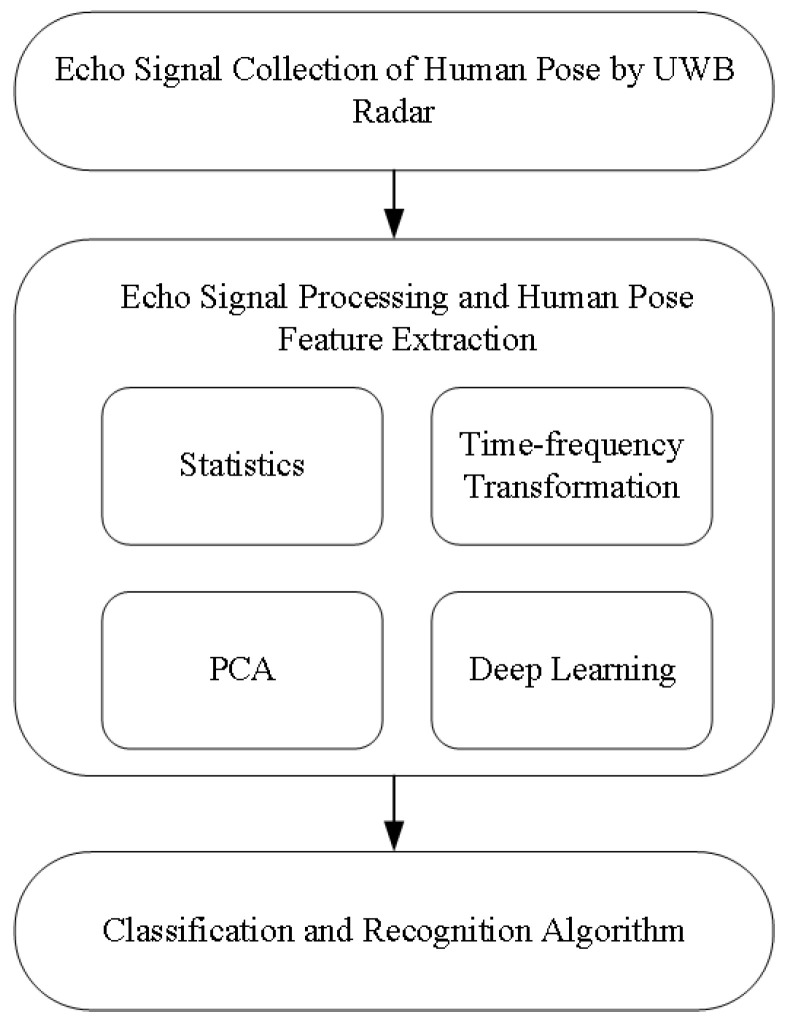
Human pose recognition model for the UWB radar.

**Figure 2 sensors-21-07130-f002:**
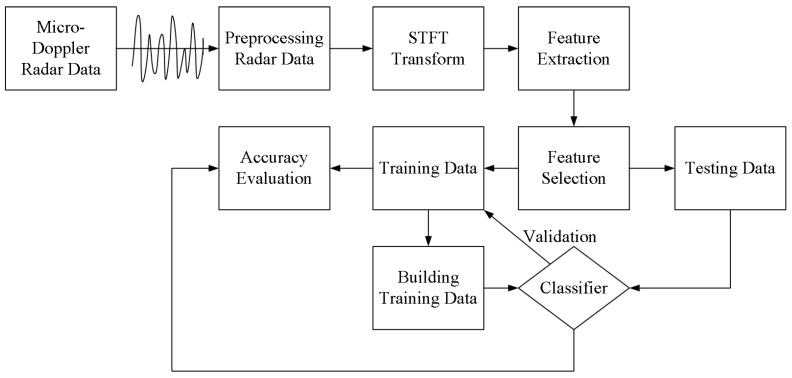
The structure of human pose classification and evaluation.

**Figure 3 sensors-21-07130-f003:**
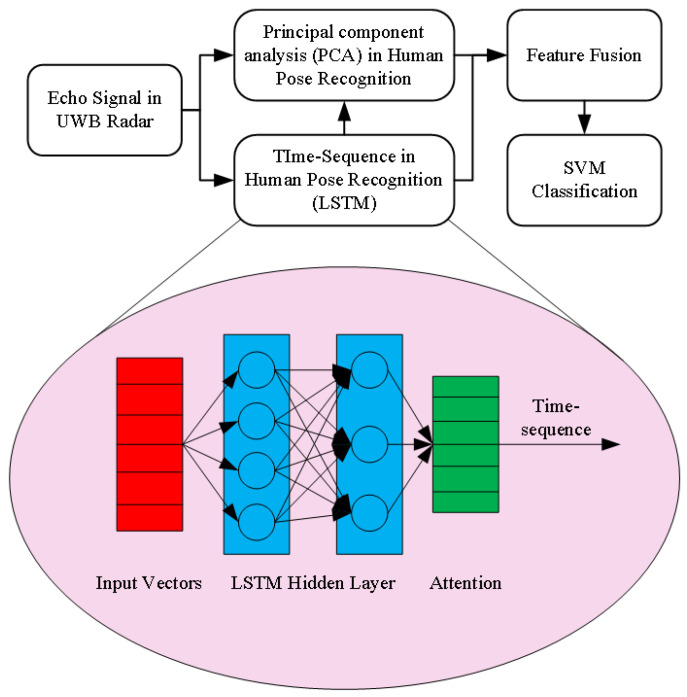
PCA-LSTM dual-channel human pose feature extraction.

**Figure 4 sensors-21-07130-f004:**
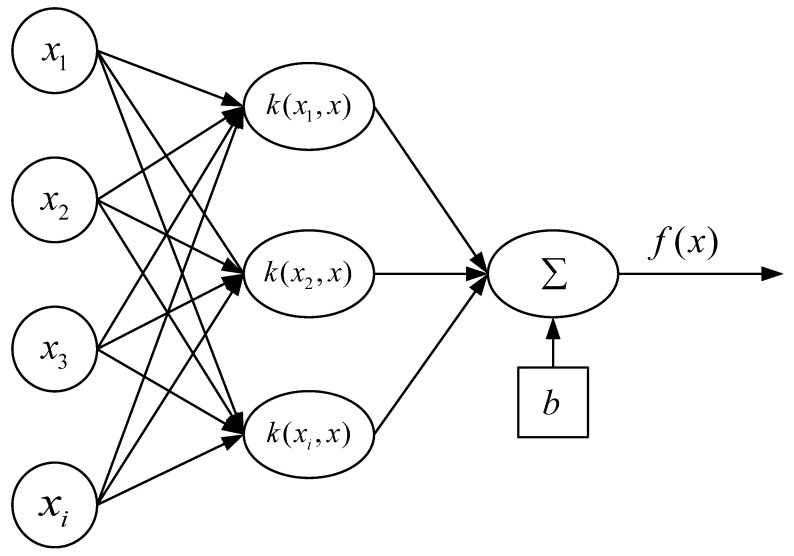
The SVM classification structure.

**Figure 5 sensors-21-07130-f005:**
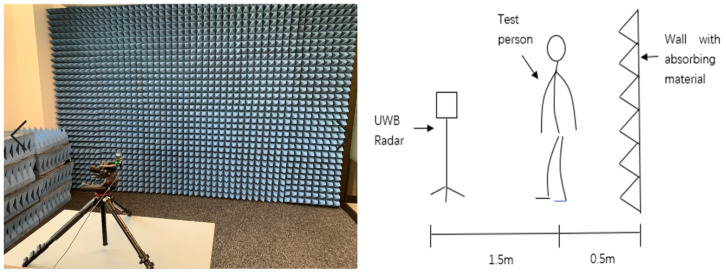
Schematic diagram of radar setting in the experimental environment.

**Figure 6 sensors-21-07130-f006:**
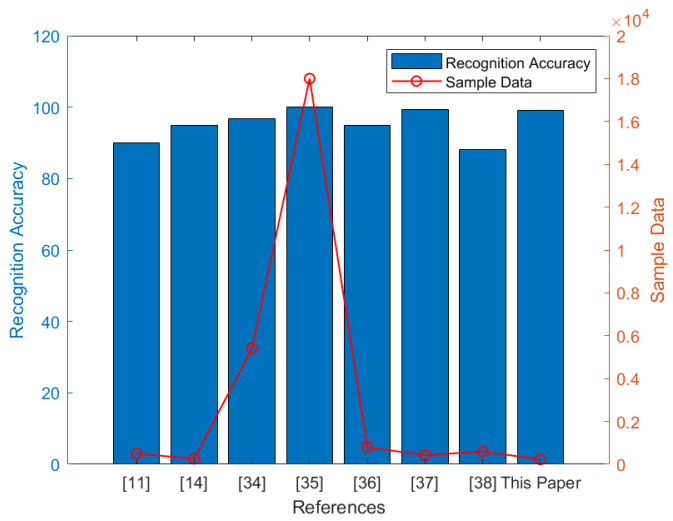
Schematic diagram in human pose recognition analysis [34,35,36,37,38].

**Figure 7 sensors-21-07130-f007:**
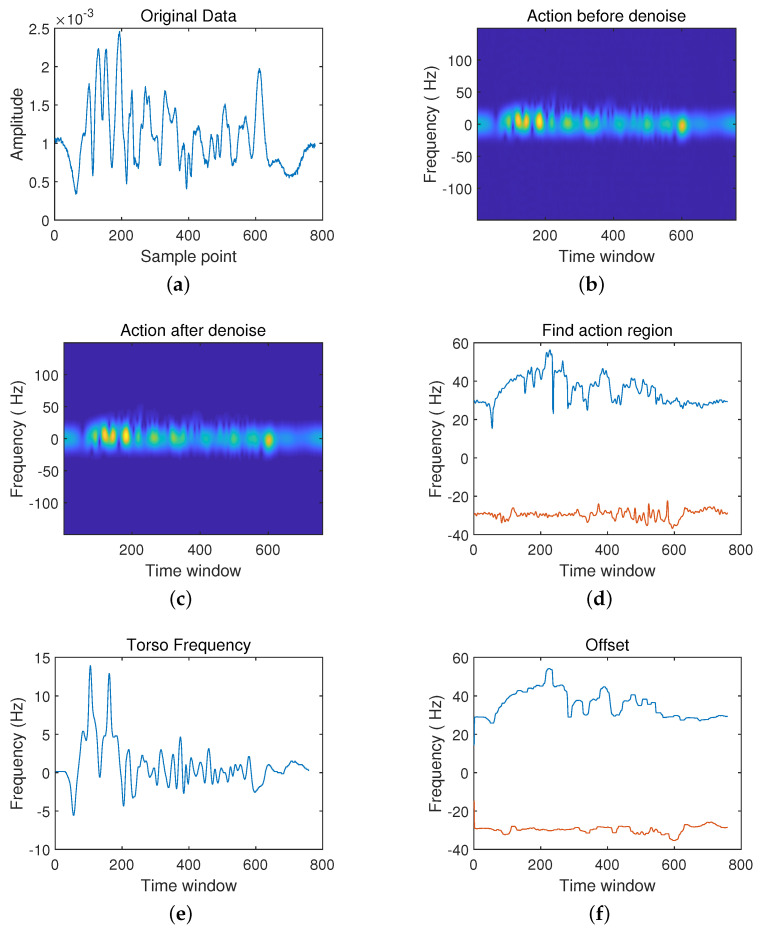
Schematic diagram from sitting to standing. (**a**) The original data.; (**b**) Frequency spectrum diagram before noise reduction; (**c**) Frequency spectrum diagram after noise reduction; (**d**) Action interval; (**e**) Body frequency; (**f**) Compensation.

**Figure 8 sensors-21-07130-f008:**
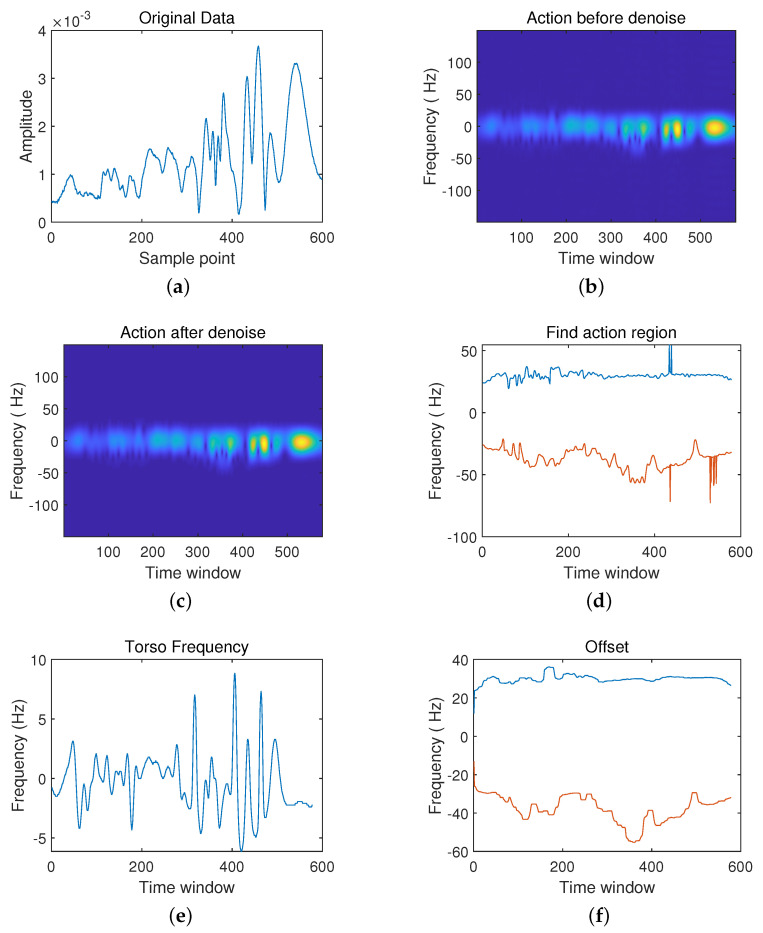
Schematic Diagram. (**a**) The original data; (**b**) Frequency Spectrum Diagram Before Noise Reduction; (**c**) Frequency spectrum diagram after noise reduction; (**d**) Action interval; (**e**) Body frequency; (**f**) Compensation.

**Figure 9 sensors-21-07130-f009:**
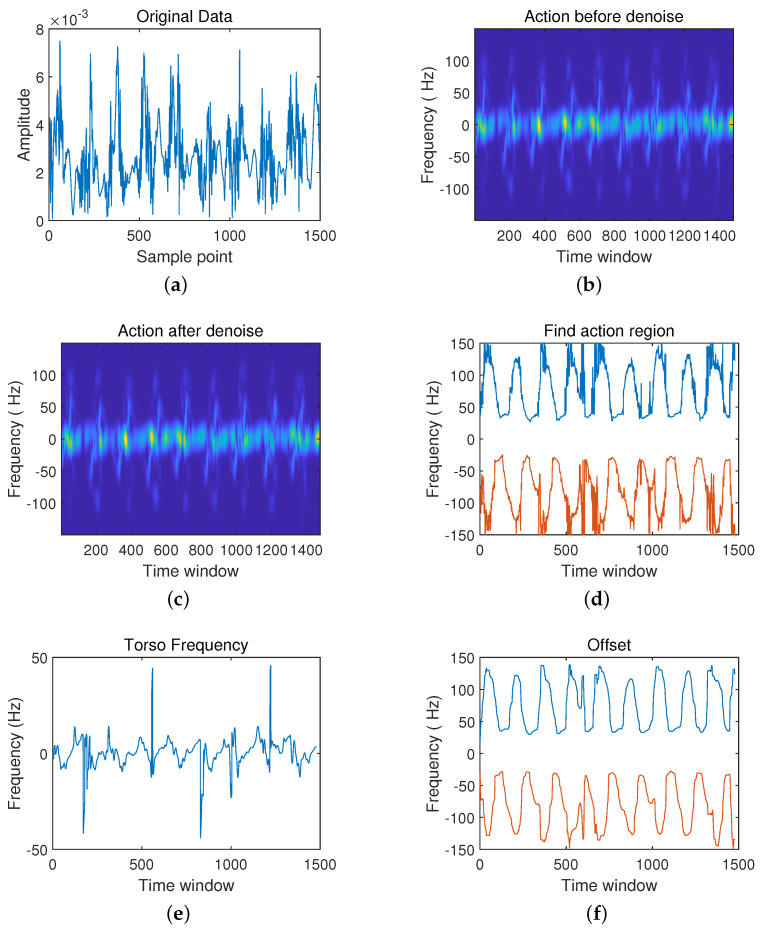
Schematic Diagram in walking in place. (**a**) The original data; (**b**) Frequency spectrum diagram before noise reduction; (**c**) Frequency spectrum diagram after noise reduction; (**d**) Action interval; (**e**) Body frequency; (**f**) Compensation.

**Figure 10 sensors-21-07130-f010:**
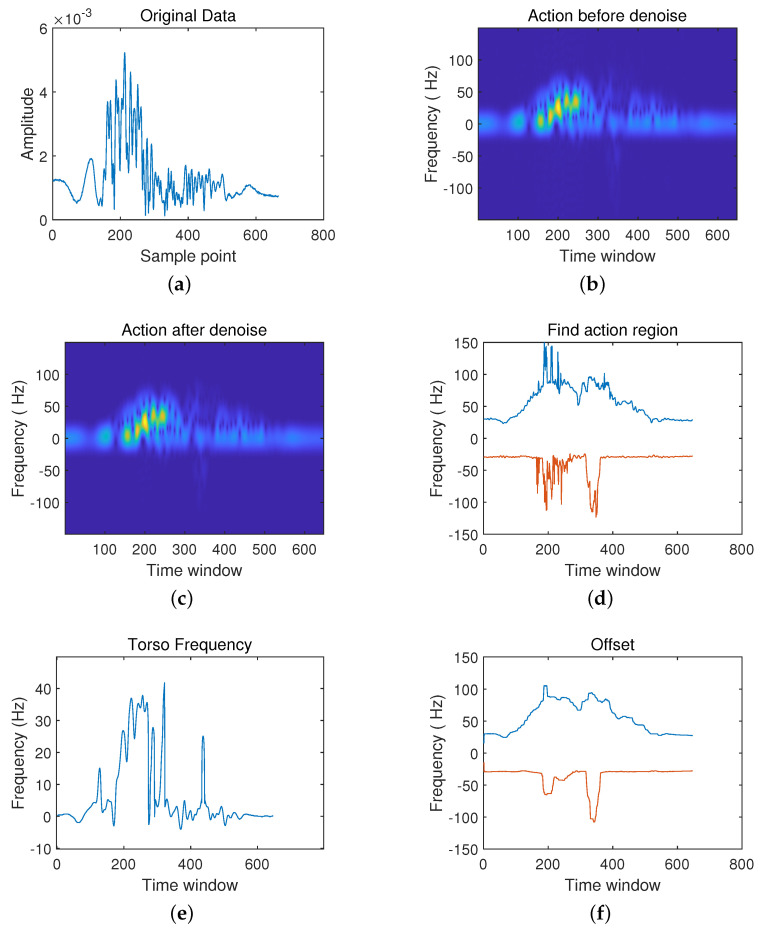
Schematic Diagram in falling. (**a**) The original data; (**b**) Frequency spectrum diagram before noise reduction; (**c**) Frequency spectrum diagram after noise reduction; (**d**) Action interval; (**e**) Body frequency; (**f**) Compensation.

**Figure 11 sensors-21-07130-f011:**
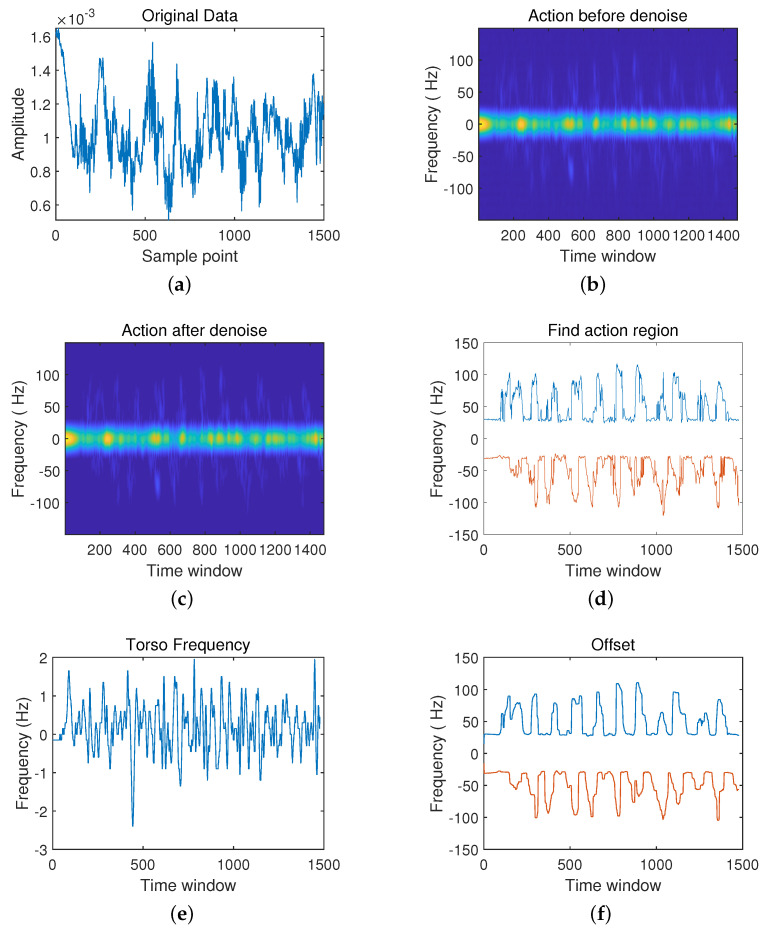
Schematic Diagram in boxing. (**a**) The original data; (**b**) Frequency Spectrum Diagram Before Noise Reduction; (**c**) Frequency spectrum diagram after noise reduction; (**d**) Action interval; (**e**) Body frequency; (**f**) Compensation.

**Figure 12 sensors-21-07130-f012:**
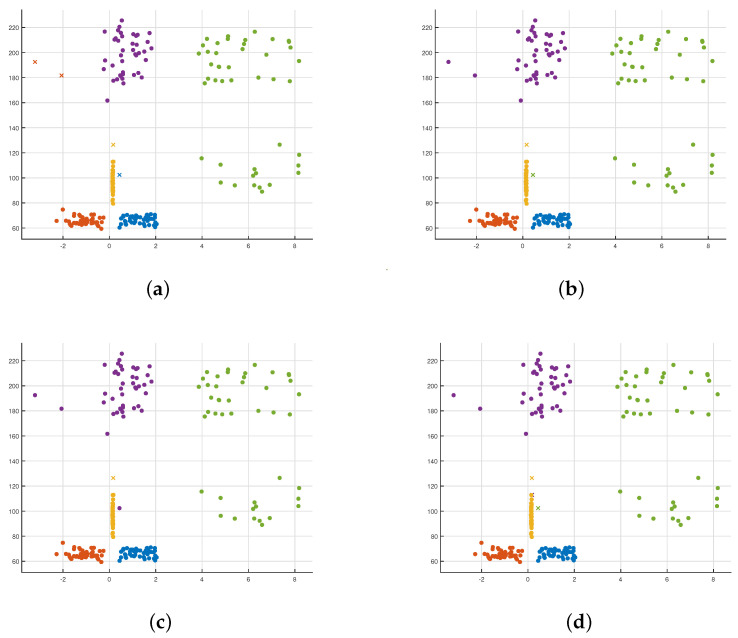
Classification models in Decision Tree, Gaussian Bayes, Gaussian SVM and KNN. (**a**) Decision Tree; (**b**) Gaussian Naive Bayes; (**c**) Gaussian SVM; (**d**) KNN.

**Figure 13 sensors-21-07130-f013:**
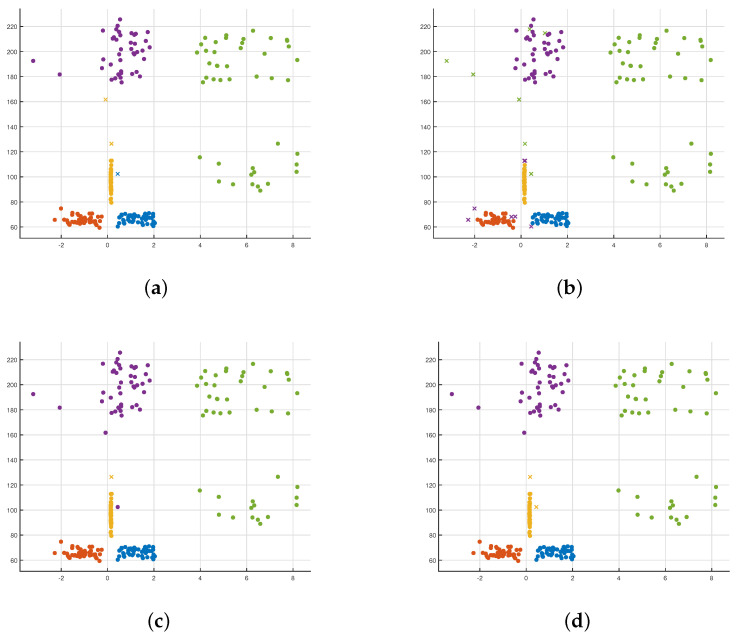
Classification models in SVM with a different kernel scale. (**a**) Linear SVM; (**b**) Gaussian σ=0.61 (SVM); (**c**) Gaussian σ=2.40 (SVM); (**d**) Gaussian σ=9.80 (SVM).

**Table 1 sensors-21-07130-t001:** Kernel function in SVM.

Kernel Function	Expression	Limitation
Linearity	$k\left(x,x_{i}\right)=\left(x,x_{i}\right)$	
polymerization	$k\left(x,x_{i}\right)={\left(x\cdot x_{i}+1\right)}^{\sigma }$	σ>0
Gaussian	$k\left(x_{i},x\right)=e^{-\frac{\gamma {∥x_{i}-x∥}^{2}}{2\sigma^{2}}}$	γ,σ>0
Laplace	$k\left(x_{i},x\right)=e^{-\frac{\gamma {∥x_{i}-x∥}^{2}}{\sigma }}$	γ,σ>0
Sigmoid	$k\left(x_{i},x\right)=than\left(\beta x\cdot x_{i}+\gamma \right)$	β,γ∈R

**Table 2 sensors-21-07130-t002:** Classification Comparison in Decision Tree, Gaussian Bayes, Gaussian SVM and KNN.

Classification	Decision Tree	Gaussian Naive Bayes	Gaussian SVM	KNN
Accuracy (%)	98.2	98.7	99.1	80.4

**Table 3 sensors-21-07130-t003:** Classification comparison in SVM with Different Kernel Scale.

SVM Kernel Function	Accuracy (%)	Training Time (s)
Linear SVM	98.7	6.4463
σ=0.61	93.8	6.0121
σ=2.40	99.1	7.4784
σ=9.80	97.1	7.3990

## Data Availability

Data Sharing not applicable.

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
