# Peer review of "Wise Information Technology of Med: Human Pose Recognition in Elderly Care"

_sensors, 2021, doi:10.3390/s21217130_

Round 1
Reviewer 1 Report
Hereafter some suggestions:
Abstract
- The abstract does not report the most relevant results (numbers), such as the accuracy and robustness of the system to sensor perturbation. It also does not indicate the amount of poses it is able to classify, as well as the type and number of people involved. This data is very important for Machine Learning based work, because it allows readers to quickly understand your work and how well it fits their purposes.
Paragraph 2:
- Please not only report the different works but, when possible, please compare the results with your approach
- You could provide some references for each method, for example for Knn you could cite "Hämäläinen, M., Mucchi, L., Caputo, S., Biotti, L., Ciani, L., Marabissi, D., & Patrizi, G. (2021). Ultra-Wideband Radar-Based Indoor Activity Monitoring for Elderly Care. Sensors, 21(9), 3158.". For Random Forest you could cite "Mizumoto, T., Fornaser, A., Suwa, H., Yasumoto, K., & De Cecco, M. (2018, April). Kinect-based micro-behavior sensing system for learning the smart assistance with human subjects inside their homes. In 2018 Workshop on Metrology for Industry 4.0 and IoT (pp. 1-6). IEEE.
Paragraph 3.
- Add privacy friendly wrt to vision based systems
- STFT have some drawbacks wrt WT, please analyse "Zhang, Y., Guo, Z., Wang, W., He, S., Lee, T., & Loew, M. (2003). A comparison of the wavelet and short-time fourier transforms for Doppler spectral analysis. Medical engineering & physics, 25(7), 547-557." and justify your use of STFT
- relative pose between the radar and the subject could have a strong inpact on the results, so please:
- specify the experimental protocol in terms of relative pose sensor/subjects
- quantify the effect of relative pose, if any with specific experimental protocol
Paragraph 5.1
- what about the classification performances in real domestic scenarios?
Paragraph 5.1
- How many people were involved?
- how big is the training dataset?
the testing dataset? - A picture of the set up with the person moving in front of the camera is needed to understand better where he/she locates with respect to the UWB sensor, how far, etc
Paragraph 5.3
-
how many people has each group?
Conclusions
- quantify results (considering similar gestures, i.e. gesture sets and environmental conditions) with respect to the state of the art
- Useful informations: People involved (helthy subjects, young or old?, how many?)
System limitations, what does affect (mainly ) the classification?
Is the system and the model robust to perturbation and repetability of data? which kind of perturbations ?
What influences the most? Does it work in any kind of room or just with the set up you tested? What may be a future improvement?
Author Response
Dear Reviewers:
Thank you for your comments concerning our manuscript entitled “Wise Information Technology of Med: Human Pose Recognition in Elderly Care” (ID: sensors-1385124). Those comments are all valuable and very helpful for revising and improving our paper. The following comments are carefully made a correction hoping to meet with approval. The major corrections in the paper and the responses to your comments are shown in PDF file.

Reviewer 2 Report
I think that this manuscript has been submitted too early. It is very difficult to follow a manuscript when the reference list includes no numbers. I found 25 references in the unnumbered list but there is a ref to 37 in one of the figures??
Further there are several sections that completely lack references and it is unclear why this solution was proposed over others. I think that a lot of information was provided in the wrong order. It surprises me that the UWB solution is proposed instead of cameras when wearable sensors could be used for activity recognition.
Further, there is a confusion about the number of movements, they are six sometimes, and five in the conclusion. But they are from sitting to standing, from standing to sitting, from walking to falling, and boxing. How are they representative for an elderly environment and what is important to monitor there?
Author Response

(The authors gave the same response as above.)

Reviewer 3 Report
Summary:
This paper proposes a human pose recognition method for elderly care. It uses UWB radar sensors for sensing, PCA-LSTM for the features and recognition processing, and SVM with different kernels for the classification problem. The authors also conducted experiments using the UWB radar with one subject.
Detailed Comments:
1. The train-test-validation dataset split is not clear. If the paper focuses on a customized solution for human pose recognition, the authors can split the data into 60%, 20%, 20% for training, testing, and validation. For example, for a 100s movement, the first 60s can be for training, the next 20s for testing, and the next 20s for validation. Or, if the authors aim at a general model, they should employ the leave-one-subject-out cross-validation setting. There should be a section introducing the training details.
2. The environment and experimental setup are not complete. This paper targets the human pose recognition problem. There is only one subject performing the movements, and it is unclear how long the whole dataset is. In my opinion, at least two subjects with 30 mins data are needed to make the results more convincing. Also, in real life, usually will not be the absorbing materials to isolate the environmental noise. Maybe the users want to do the same experiment with and without the noise absorbing materials. The result can be pretty interesting. Also, it is more meaningful to the real problem. The authors can also consider playing with the distance between the wall and the subjects since they claim UWB sensors' sensing distance is long, yet they only perform the experiment in the 1.5m range.
3. The explanation is confusing. There are many examples. Line 321: "Let define 0 to 5 be standing, sitting, walking in place, failing, and boxing, respectively. " There are only five poses here, but the sentence indicates that there are six by the "0 to 5" statement. This conflict exists in many places.
Line 329: "However, in Fig. 12(a), there are two fork points of yellow and green." It should be yellow and blue.
Line 321: "... failing, and boxing, respectively". It should be "falling". This typo exists in multiple places.
Also, there should be tables for the accuracy result instead of only figures 12 and 13.Summary:
4. Recent studies consider human pose estimation. For example, one study proposes a technique for human pose estimation, including 19 critical joints 3D positions, using the mmWave radar [1]. It is also related to the elderly care/rehabilitation movement. There are also papers working on reconstructing human key points [2] and human mesh [3] from radar in general, not only for the elderly care/rehabilitation. The authors may want to compare with these papers (at least qualitatively) and consider estimating joints positions, which offer more information, instead of only doing pose recognition. If it is not applicable for the UWB sensors, the authors should mention them in the related work and claim why it is the case.
[1] Sizhe An and Umit Y. Ogras. 2021. MARS: mmWave-based Assistive Rehabilitation System for Smart Healthcare. ACM Trans. Embed. Comput. Syst. 20, 5s, Article 72, 22 pages.
[2]Sengupta A, Jin F, Zhang R, et al. mm-Pose: Real-time human skeletal posture estimation using mmWave radars and CNNs[J]. IEEE Sensors Journal, 2020, 20(17): 10032-10044.
[3]Xue H, Ju Y, Miao C, et al. mmMesh: towards 3D real-time dynamic human mesh construction using millimeter-wave[C]//Proceedings of the 19th Annual International Conference on Mobile Systems, Applications, and Services. 2021: 269-282.
Author Response

(The authors gave the same response as above.)

Round 2
Reviewer 1 Report
Thanks for considering the listed comments.
Reviewer 3 Report
The authors addressed our comments.